# Causal Hierarchy in the Financial Market Network—Uncovered by the Helmholtz–Hodge–Kodaira Decomposition

**DOI:** 10.3390/e26100858

**Published:** 2024-10-11

**Authors:** Tobias Wand, Oliver Kamps, Hiroshi Iyetomi

**Affiliations:** 1Institute of Theoretical Physics, University of Münster, Wilhelm-Klemm-Straße 9, 48149 Münster, Germany; 2Center for Nonlinear Science, University of Münster, Corrensstr. 2, 48149 Münster, Germany; okamp@uni-muenster.de; 3Faculty of Data Science, Rissho University, 1700 Magechi, Kumagaya 360-0194, Japan; hiyetomi@ris.ac.jp; 4Canon Institute for Global Studies, 5-1 Marunouchi 1-chome, Chiyoda-ku, Tokyo 100-6511, Japan

**Keywords:** financial networks, Granger causality, Helmholtz–Hodge, econophysics, causal inference

## Abstract

Granger causality can uncover the cause-and-effect relationships in financial networks. However, such networks can be convoluted and difficult to interpret, but the Helmholtz–Hodge–Kodaira decomposition can split them into rotational and gradient components which reveal the hierarchy of the Granger causality flow. Using Kenneth French’s business sector return time series, it is revealed that during the COVID crisis, precious metals and pharmaceutical products were causal drivers of the financial network. Moreover, the estimated Granger causality network shows a high connectivity during the crisis, which means that the research presented here can be especially useful for understanding crises in the market better by revealing the dominant drivers of crisis dynamics.

## 1. Introduction

One of the most important messages in many introductory lectures to statistics is that correlation does not imply causation [1]. However, this begs the following question: What, then, is causality? And how can it be quantified? One of the first and most widespread attempts to formalize causality was proposed by Granger [2]. Granger causality takes the time ordering into account, as the cause needs to happen before the effect. The field of causal inference has developed several tools to probe time series data for causal interactions [3], and it has been used to analyze dynamical systems [4,5]. Especially for high-dimensional multivariate time series, it is difficult to infer the network of causality because one has to carefully distinguish between the different possible causal drivers [6,7,8,9].

While such networks can be convoluted and difficult to interpret, especially if they contain cyclic substructures [10], the Helmholtz–Hodge–Kodaira decomposition (HHKD) can disentangle them. As a reformulation of Helmholtz–Hodge field theory for discrete graphs, the HHKD can split a directed network into a cyclic graph and a gradient-based graph [11,12]. The latter will then provide a ranking of all nodes according to whether they are upstream or downstream. Cyclic substructures can pose a problem in the frameworks of causal inference [3], and the HHKD’s ability to split them apart from the main network can help to interpret the causal flux between the nodes. The remaining gradient-based flux will then provide a hierarchical ranking of the causal drivers. The application of methods and tools from physics to economic and financial systems is known as econophysics [13], and the HHKD has been used in this field to understand the dynamics of cryptocurrencies [14], as well as the networks of shared ownership of companies [15]. Capturing economic and financial interactions in a network has long been a standard approach within econophysics and complexity science [16,17,18], and causal inference has been applied to such networks of companies or countries [19,20,21].

This article analyzes time series data on business sectors from [22] to investigate the Granger causality between different sectors of the economy. Using the HHKD on this Granger network then reveals whether a sector is driven by other sectors or rather whether it is a causal driver of the whole system. First, this article will present the database, the algorithm from [8] for estimating Granger causality networks, and the HHKD for the graphs in Section 2. The results of the HHKD will then be presented for different time periods in Section 3 before they are interpreted and further extensions to this research are discussed in Section 4.

## 2. Materials and Methods

### 2.1. Data

To analyze the interactions between different sectors of the economy, we use the database of Ken French, which contains return time series of representative portfolios for 49 different business sectors [22]. These portfolios are constructed as value-weighted averages of all the stocks in a business sector listed on NYSE, AMEX, and NASDAQ, and the data consist of the daily returns $R_{t}^{\left(i\right)}=\left(P_{t}^{\left(i\right)}-P_{t-1}^{\left(i\right)}\right)/P_{t-1}^{\left(i\right)}$ of these portfolios’ prices. The calculation of returns normalizes the time series and removes trends in the data. Moreover, augmented Dickey–Fuller tests [23] performed via the Python package *statsmodels* [24] indicate that the given time series are stationary for each window period of interest in Section 3, with the exception of one sector each in 2006 and 2008 (though that does not impact our results). This database is updated continuously with further details on the data curation given in [25]. Although the assignment of companies into sectors was conducted manually, a comparative study with modern statistical tools shows high agreement between French’s classification and data-driven methods [26]. While the data were originally used for capital asset pricing modeling in [27], they have found numerous applications in various fields of economic and financial research as a data resource (see [26] for an overview).

### 2.2. Granger Causality

The intuition behind Granger causality is that the cause *X* should happen before the effect *Y* and that knowing the cause should improve the future prediction of the effect. The latter can be measured by fitting autoregressive linear models with and without *X* and comparing their accuracy. By including possible alternative causes *Z* for *Y*, this concept is extended to the conditional Granger causality, *CGC*. First, a full model is estimated that measures how well the past of *X*, *Y*, and the background variables *Z* predict the future of $Y_{t+1}$ via
(1)$$Y_{t+1}=\sum_{\tau =0}^{\tau_{\max}}\left(\alpha_{\tau }Y_{t-\tau }+\beta_{\tau }X_{t-\tau }+\gamma_{\tau }Z_{t-\tau }\right)+\epsilon$$
with i.i.d. Gaussian errors $\epsilon ∼N(0,\sigma_{F}^{2})$ and a maximum time lag $\tau_{\max}$ to limit how much of the past should be considered in predicting $Y_{t+1}$. Note that *Z* might contain more than one background variable $Z=\left(Z^{\left(1\right)},\ldots ,Z^{\left(s\right)}\right)$ with $\gamma_{\tau }\in R^{s}$. Then, a reduced model is trained without the proposed cause *X* as
(2)$$Y_{t+1}=\sum_{\tau =0}^{\tau_{\max}}\left(\alpha_{\tau }^{\prime }Y_{t-\tau }+\gamma_{\tau }^{\prime }Z_{t-\tau }\right)+\epsilon$$
with $\epsilon ∼N(0,\sigma_{R}^{2})$. The conditional Granger causality of *X* on *Y* is then given by how much the reduced model’s variance increases compared to the full model and is defined as
(3)$$CGC_{X\to Y}=\log\frac{\sigma_{R}^{2}}{\sigma_{F}^{2}}$$
to measure how much *X* causes *Y*.

For multivariate data with many time series, the estimation of the full model in (Equation 1) can easily fall into the regime of overfitting [28]. Hence, it is of paramount importance to construct the full model carefully. A comparative study of multivariate Granger networks [9] indicates that the restricted conditional Granger causality index (RCGCI) from [8] is the most suitable Granger causality estimation scheme for the financial data analyzed in this article. At the heart of the RCGCI lies the construction of a full model by starting with an empty regression model and sequentially adding variables $X_{t-k\tau }^{\left(i\right)}$ with a lag of *k* time units to it if they reduce the *BIC* of the regression [29]. Hence, the resulting full model may not contain lagged representations of all possible explanatory variables but only some selected ones $\left(X^{\left(i_{1}\right)},ߪ,X^{\left(i_{I}\right)}\right)$, and this therefore guarantees sparsity to prevent overfitting in the estimation process. For these variables, *CGC* can be computed by removing them from the full model and fitting the reduced model, whereas the remaining selected variables are conditioned on as the background information *Z*. For the other $X_{j}$ values which have not been included in the full model, the *CGC* is set to zero, as no causal relationship had been estimated.

#### Details on the Estimation

Because the financial returns analyzed in this article are known to have an almost nonexistent autocorrelation [13] and to avoid overfitting, we restrict our models to maximum lags of one time step. As this represents a full day of trading activity, data with a lag of two time units (i.e., two days), in the fast-paced and constantly adapting environment of financial markets, yield little additional contribution to the full Granger model, as shown by an exploratory analysis. Previous studies have shown that principal component analysis (PCA) can be used to distinguish between noise and collective effects in financial time series [30,31]. Hence, we perform PCA on the raw data, only keep the principal components with the largest eigenvalues so that their sum describes 90% of the total variation in the data, and discard the remaining principal components as noise before performing the inverse transformation back into the original feature space. Note that the sparsity of the RCGCI algorithm also limits the influence of noise on the results. Averaging over all sectors and all time periods under consideration, the typical ratio between the variance explained by the full regression model and the variance in the data is $\sigma_{F}^{2}/\sigma_{\mathrm{Data}}^{2}\approx 96\%$, indicating the good fit of the full regression model and a high signal-to-noise ratio.

### 2.3. Helmholtz–Hodge–Kodaira Decomposition

The reconstructed network of the causality flux between multivariate time series might not be easy to interpret ad hoc. Circular causalities (*A* causes *B*, *B* causes *C*, and *C* causes *A*) may be present, and inspecting the network with the naked eye may not be sufficient to understand its structure. The Helmholtz–Hodge–Kodaira decomposition (HHKD) is a tool for analyzing the flux in networks and disentangling the flow into upstream and downstream directions [11,12].

#### 2.3.1. Mathematical Formulation of the Unidirectional HHKD

The Helmholtz decomposition theorem states that any well-behaved vector field $F(r)\in R^{n}$ can be decomposed into two components F(r)=G(r)+R(r), a gradient field **G(r)** and a divergence-free field **R(r)**. The rotation-free field **G(r)** can be expressed as the gradient of a potential $G(r)=-\nabla_{r}\Phi (r)$ such that the potential determines the direction of a flux in the space of **r**. For the divergence-free or solenoidal field R(r), no point *r* is a source or sink of the observed flux, as $\forall r:\nabla_{r}\cdot R(r)=0$. Note that a third component may exist and represent a background flux into and out of the system, but it is usually ignored, as one assumes that the system of interest is sufficiently closed.

The same reasoning can be applied to a flow network on a discrete graph [11,12]. Let $J_{ij}$ be the observed flow from node *i* to *j* with the antisymmetric property $J_{ij}=-J_{ji}$. It can be shown that a unique decomposition $J_{ij}=J_{ij}^{\left(g\right)}+J_{ij}^{\left(c\right)}$ exists such that $J_{ij}^{\left(g\right)}$ is the gradient and $J_{ij}^{\left(c\right)}$ is the circular flow from *i* to *j*. In this decomposition, the gradient flux fulfills $J_{ij}^{\left(g\right)}=G_{ij}\left(\Phi_{i}-\Phi_{j}\right)$ for some background potential Φ assigned to each node and with the standard choice for the weights between two nodes being $G_{ij}=1$. The circular flow fulfills $\forall i:\sum_{j}J_{ij}^{\left(c\right)}\overset{!}{=}0$, i.e., for each node, the total influx is equal to its total outflow. For a simple network with three nodes, this decomposition is illustrated in Figure 1. The potential and its associated gradient flow can be obtained from the least square estimation
(4)$$\min_{J^{\left(g\right)}}\left(I\right)\;\mathrm{with}\;I=\frac{1}{2}\sum_{i<j}\frac{1}{G_{ij}}{\left(J_{ij}-J_{ij}^{\left(g\right)}\right)}^{2}=\frac{1}{2}\sum_{i<j}\frac{1}{G_{ij}}{\left(J_{ij}-G_{ij}\left(\Phi_{i}-\Phi_{j}\right)\right)}^{2}$$
and the circular flow is then simply the difference $J_{ij}^{\left(c\right)}=J_{ij}-J_{ij}^{\left(g\right)}$. For the standard choice $G_{ij}=1$, this formulation also has the useful property that the net gradient flux is the same along all paths between any two nodes. Because the gradient flow only depends on the potential difference $\Phi_{i}-\Phi_{j}$, the same gradient flow can also be obtained if the potentials have a constant offset $\Phi_{i}\to \Phi_{i}+\Phi_{O}$. Hence, the minimization of Equation (Equation 4) needs an additional constraint to produce unique results for Φ, such as $\Phi_{n}=0$ or $\sum_{i}\Phi_{i}=0$. We note that the mathematical formulation of the HHKD tends to minimize large errors comparatively more than small ones and treat large flows between two nodes as more informative than small ones. This means that any necessary errors during the minimization procedure are distributed across many edges. Because of the sparse RCGCI algorithm, the weakest causal links are discarded before the application of the HHKD, and hence, the errors will be relatively weak compared to the strength of the estimated network links. This shortcoming should be kept in mind. We additionally note that, as proven for a trait performance model in [32], the uncertainty in the estimation of $CGC_{X\to Y}$ introduces bias towards more cyclical structures.

#### 2.3.2. Bidirectional Flows

Whether through noise or feedback loops, in general, it is possible for the RCGCI algorithm to estimate that $CGC_{X\to Y}>0$ and $CGC_{Y\to X}>0$, i.e., that two time series are estimated to Granger-cause each other and that the flux cannot be defined as antisymmetric $J_{ij}^{\left(b\right)}\neq -J_{ji}^{\left(b\right)}$ where the superscript *b* denotes the bidirectionality and $J_{ij}^{\left(b\right)}\geq 0$. Naively, computing the net flow $CGC_{X\to Y}-CGC_{Y\to X}$ seems like a reasonable choice, but this discards the information about the relative strength of the net flux. Consider a system in which $CGC_{A\to B}=0.6$, $CGC_{B\to A}=0.1$, $CGC_{C\to D}=5.5$, and $CGC_{D\to C}=5$. The net flux A→B, and C→D is 0.5, but scaling this with the total flux between the node pairs shows that this difference is much less significant for the flux C→D.

A bidirectional version of the HHKD that reflects these considerations is presented in [15]. The authors argue for interpreting $G_{ij}$ as an analogy for the conductance in electrical circuits, where a high flux between both nodes corresponds to high conductance, whereas a low total flux indicates high resistance. Hence, they propose splitting the original bidirectional network into two graphs, which are then used to perform the HHKD: the difference in the flux in both directions between two nodes in the bidirectional network $J^{\left(b\right)}$ is defined as the net flux $J_{ij}J_{ij}^{\left(b\right)}-J_{ji}^{\left(b\right)}$ and forms a unidirectional network with the antisymmetry $J_{ij}=-J_{ji}$ to which the HHKD can be applied. The sum of the absolute values of the flux in both directions is used as the conductivity $G_{ij}J_{ij}^{\left(b\right)}+J_{ji}^{\left(b\right)}$, which is the same in both directions $G_{ij}=G_{ji}$. The minimization in Equation (Equation 4) can then be applied to the two networks (J,G) to receive the HHKD ranking of the nodes.

This generalization also gives rise to a helpful interpretation of the potential differences in the nodes: consider only the flux between two nodes *i* and *j* isolated from the rest of the graph. Let $J_{ij}$ be the net flow from *i* to *j* and $G_{ij}$ the total flow. The contribution of this minimized connection to the functional *I* is then given by
(5)$$I_{ij}=\frac{1}{2}\frac{1}{G_{ij}}{\left(J_{ij}-J_{ij}^{\left(g\right)}\right)}^{2}=\frac{1}{2}\frac{1}{G_{ij}}{\left(J_{ij}-G_{ij}\left(\Phi_{i}-\Phi_{j}\right)\right)}^{2}=\frac{1}{2}\frac{1}{G_{ij}}{\left(J_{ij}-G_{ij}\Delta_{ij}\right)}^{2}$$
where $\Delta_{ij}$ expresses the potential difference between the two nodes. Minimization of $I_{ij}$ with respect to $\Delta_{ij}$ leads to
(6)$$\frac{\partial I_{ij}}{\partial \Delta_{ij}}=-J_{ij}+G_{ij}\Delta_{ij}\overset{!}{=}0\Leftrightarrow \Delta_{ij}=\frac{J_{ij}}{G_{ij}}\equiv \frac{\mathrm{Net}\;\mathrm{Flow}}{\mathrm{Total}\;\mathrm{Flow}}.$$
Hence, this rule-of-thumb approximation, which disregards all other edges, shows that the potential difference can be interpreted as the ratio between the net and total flow between the two nodes. In particular, if the flux in one direction is much larger than in the other direction $J_{ij}^{\left(b\right)}\gg J_{ji}^{\left(b\right)}$, then the net and total flow are almost identical $J_{ij}\approx G_{ij}$, so that $\Delta_{ij}\approx 1$. Therefore, as described in [15], one unit of potential difference can be interpreted as a separation of approximately one layer between the nodes *i* and *j*.

#### 2.3.3. Circularity and Hierarchy

Once the flow network is decomposed into gradient-based and circular flux, one can compare their respective contributions to the net flux [15,33,34]. It is possible to quantify the contribution of the gradient-based flux via the $L^{2}$ norm as
(7)$$\Gamma =\frac{1}{2}\sum_{i}\Gamma_{i}=\frac{1}{2}\sum_{i}\sum_{j}G_{ij}{\left(J_{ij}^{\left(g\right)}\right)}^{2}$$
and that of the circular flux as
(8)$$\Lambda =\frac{1}{2}\sum_{i}\Lambda_{i}=\frac{1}{2}\sum_{i}\sum_{j}G_{ij}{\left(J_{ij}^{\left(c\right)}\right)}^{2}$$
where $\Gamma_{i}$ and $\Lambda_{i}$ denote the contribution of the respective $i^{\mathrm{th}}$ node. Normalizing them with the total flux
(9)$$N=\frac{1}{2}\sum_{i}\sum_{j}G_{ij}{\left(J_{ij}\right)}^{2}$$
leads to the definition of
(10)$$\gamma =\frac{\Gamma }{N}\;\mathrm{and}\;\lambda =\frac{\Lambda }{N}$$
which fulfill γ+λ=1. In a completely hierarchical network, γ=1, while in a completely circular network, λ=1. A high γ≫λ indicates that the underlying potential and its corresponding hierarchy have been cleansed of noise and insignificant loops and now accurately reflect the true structure of the underlying dynamics.

### 2.4. Test on Synthetic Data

To test the pipeline of the RCGCI and HHKD, we simulated 50 realizations of a network of 49 time series with the network structure given in Figure 2. Because the RCGCI-HHKD pipeline is supposed to uncover the hierarchy of time series, the synthetic network in Figure 2 was chosen to represent a hierarchical structure, and we evaluated how accurately the algorithm identified the nodes at the top of the hierarchy: one node $X_{0}$ is an independent stochastic process and is at the top of the hierarchy. It drives the nodes in the next layer $X_{1,\ldots ,8}$ as their causal parent pa, and each of them drives five nodes from $X_{9,\ldots ,48}$ in the final layer at the bottom of the hierarchy. Each node $X^{\left(i\right)}$ in the network was simulated for 250 time steps in vector autoregression according to
(11)$$X_{t+1}^{\left(i\right)}=-0.5X_{t}^{\left(i\right)}-0.5X_{t}^{pa(i)}+\epsilon$$
where pa(i) denotes the parent node of *i* (if it exists) and $\epsilon \overset{iid}{∼}N\left(0,\sigma \right)$ with σ=1. The goal of the RCGCI-HHKD pipeline is to accurately identify the nodes with index I=(0,…,8) at the top of the causal hierarchy. Note that 49 time series for 250 time steps correspond to one year of trading day data in the database described in Section 2.1 and that the standard deviation σ=1 is roughly equal to the standard deviation of the data. Hence, these synthetic time series provide a realistic artificial version of the observed data. The HHKD is used on the Granger causality network estimated by the RCGCI, and we evaluate how many of the top nine nodes in the estimated hierarchy actually belong to the set I. For the 50 realizations of the network, the average of the ensemble of detection rates is 94%, and the median is 100%. Though this is not shown here, for synthetic cyclic networks, the RCGCI also successfully estimates the loop topology. Hence, the combination of the RCGCI and HHKD accurately estimates the network structure. Even after adding observation noise $N(0,\sigma_{\mathrm{obs}})$ to the simulated data via
(12)$$\begin{array}{c} X_{t}^{\left(i\right)}\to {\tilde{X}}_{t}^{\left(i\right)}=X_{t}^{\left(i\right)}+\zeta \;\mathrm{with}\;\zeta ∼N\left(0,\sigma_{\mathrm{obs}}\right), \end{array}$$
the top nodes in the network hierarchy are still estimated with a high accuracy beyond random expectations for noise up to $\sigma_{\mathrm{obs}}\approx 1$.

Additionally, we also create ensembles of uncoupled time series by taking a random subset of trading days $t_{i_{1}},\ldots ,t_{i_{250}}$ without any chronological ordering. This represents the null hypothesis that no causal coupling is present in the data. We then check whether the RCGCI erroneously reconstructs a network even though none is present. We define the network connectivity as the percentage of nodes that are estimated to have a causal coupling to another node. These 50 percentage values give us a confidence interval (CI) of the network connectivity under the null hypothesis that no causal coupling is present in the data. If the network connectivity of a system exceeds the CI, we can therefore conclude that we have identified a system with meaningful causal connections.

## 3. Results

### 3.1. Year by Year

To gain the most insight from the HHKD, it is necessary to have a connected network in which all sectors are included. By defining the network connectivity as the percentage of nodes which are coupled to the network, the RCGCI-HHKD analysis is performed on the annual data for the last 20 years from 2004 to 2023 to identify periods with a connectivity of 100%.

The results in Figure 3 show that the market mostly remains within the range expected of random time series, but some periods exhibit a spike to a significantly high level of network connectivity. It only reaches a connectivity of 100% (i.e., with all sectors coupled to the network) during the year 2020 and reaches a connectivity of almost 100% (with only one sector decoupled) in 2007. Hence, the 2020 period will be the focus of the latter half of this section.

To gain more insights into the general structures of the RCGCI networks across the years, the sum of the Granger causality influx and outflux is recorded for each of the 49 sectors, as well as for how many years they are connected to the network by influx or outflux links. After rescaling all of these quantities to the same scale, kernel density estimation (KDE; [36]) in Figure 4 shows that for the inward and outward directions, the total flux and the linkage rate are distributed similarly. The influx distribution is a Gaussian bell curve (a two-sided $\chi^{2}$ test has a p value of 0.21 and cannot reject the null hypothesis of Gaussianity), whereas the outflux has a higher variance and, notably, a fat tail at high values. Hence, while most sectors have a similar influx of Granger causality, some sectors drive the other sectors with a much stronger outward Granger causality than most of the others. It is therefore more interesting to focus on sectors with a particularly high or low outflux of Granger causality. The sectors Rubbr (rubber and plastic products), BldMt (construction materials), Mach (machinery), Trans (transportation), and, perhaps surprisingly, Banks show no outflux of Granger causality during any of the periods. Gold (precious metals) and, in the second position, Cnstr (construction) have a much higher sum of Granger causality outflux and a much higher rate of outward linkage than all the other sectors.

### 3.2. Complete Graphs during the 2020 COVID Pandemic

To investigate the connected network for the year 2020 further, the RCGCI-HHKD pipeline is used to analyze time windows of 12 months which are shifted by 1 month and scan over the year 2020. This process starts with the time interval of January 2019 to January 2020 and ends with the period from December 2020 to December 2021. Note that the figures displaying these results use the midpoint of each time period on the x-axis, e.g., July 2019 for the period from January 2019 to January 2020. For each period with a connected network, the parameter γ is calculated according to Equation (Equation 10) to quantify the contribution of the gradient flow to the observed flux. Because λ=1−γ, the calculation of λ is omitted.

Figure 5 shows the results for the connectivity and the gradient contribution γ. Whether the network is complete or not depends on whether March 2020 is included in the data, as including March 2020 seems to guarantee that the connectivity reaches 100%. During this period, the gradient contribution is typically around γ≈0.8 and therefore stronger than the rotational flow λ. However, due to the quadratic $L^{2}$ norm used to calculate (Equation 10), a rotational component λ≈0.2 is nevertheless a non-negligible contribution to the total flow.

Looking at the ensemble of Granger causality flux matrices for the 12 time windows with a connectivity of 1 reveals that every single sector always has some causality influx (i.e., it is estimated to be Granger-caused by another sector) for all 12 time windows, with the exception of the sector Gold, which only has an influx of Granger causality for 3 of the 12 periods. The sectors PerSv (personal services), Other, Aero (aircraft), and Trans (transportation) have no outflux of Granger causality during any of the 12 time windows, and for the latter two sectors, this might reflect the travel restrictions imposed during this period. In contrast to the sum of the Granger causality outflux for the disjoint long-term analysis in the previous section, Gold is no longer the sector with the highest total outflux (≈5.00) but is far overtaken by Drugs (pharmaceutical products; ≈35.8), Hshld (consumer goods; ≈28.4), and Cnstr (≈10.6), with some other sectors at a slightly higher level than Gold, too.

For the periods with complete graphs, the potential $\Phi_{i}$ can be calculated for every single node. A high $\Phi_{i}$ indicates a high position in the hierarchy of Granger causality and that the sector is a cause rather than an effect. Because of the large contribution γ of the gradient flux shown in Figure 5, this hierarchy is not obstructed by strong circular fluxes in the system and indeed reflects the underlying dynamics. Figure 6 shows the potentials for all 12 time windows with a complete graph. These potentials can be compared to each other because they have been normalized to fulfill $\sum_{i}\Phi_{i}=0$ for each time period. The range between the minimum and maximum potential values has a mean of 2.0 with a standard deviation of ±0.2 across these periods and therefore reflects a network of approximately three different levels with a fairly stable potential range. Additionally, for the period from October 2019 to September 2020, the full network is depicted in Figure 7, where the nodes’ vertical positions reflect their potential values. Some selected sectors have been highlighted in these plots: the sectors Gold and, with the exception of the first interval, Drugs are consistently at the top of the hierarchy, and their potentials have a low variance. Similarly, the potentials of the sectors Aero, Meals (Restaurants, Hotels, and Motels), and RlEst (Real Estate) have certain mean values and variances. Therefore, these sectors are consistently found at the bottom of the potential hierarchy.

Some other sectors have a high variance and change their position in the hierarchy rather drastically: the potential of Cnstr has the highest variance, and this sector moves upwards in the hierarchy during the latter third of the periods. This might reflect the increase in construction material prices and their effect on construction businesses and, as a cascading effect, on other business sectors during the beginning of 2021 [37]. The second highest variance is observed for the potential of MedEq (medical equipment). This sector rises in the hierarchy during the peak of COVID, which probably reflects the increasing demand for products such as face masks and testing equipment. The sudden decline of MedEq in the hierarchy starts in the time windows that include the first weeks of widespread vaccinations in Western countries, which were interpreted as a sign of the end of the pandemic and hence of the lower importance of such equipment.

Because of the small but notable contribution of the rotational flux to the system during the COVID crisis, we also investigate which nodes have a strong rotational component $\Lambda_{i}$, as in Equation (Equation 8). For each time period with a connected network graph, the values $\Gamma_{i}$ and $\Lambda_{i}$ are calculated according to Equations (Equation 7) and (Equation 8), and the rotational component is normalized in two ways: $\Lambda_{i}^{\left(N\right)}=\Lambda_{i}/N$ denotes how much the rotational flux of node *i* contributes to the total flux *N* from Equation (Equation 9). $\lambda_{i}=\frac{\Lambda_{i}}{\Lambda_{i}+\Gamma_{i}}$ denotes whether the flux of node *i* is dominated by rotational flows or rather by the gradient flow. For each sector *i*, the mean values of $\Lambda_{i}^{\left(N\right)}$ and $\lambda_{i}$ are calculated over the time periods of consideration. While the sectors Drugs, Hshld, and Cnstr have the highest, second highest, and fourth highest mean values of $\Lambda_{i}^{\left(N\right)}$ and contribute greatly to the rotational flow, their own flows do not show a particularly high contribution $\lambda_{i}$ of the rotational component. Rather, their $\Lambda_{i}^{\left(N\right)}$ is high because they have strong causality links to other sectors in general. The third and fifth highest values of $\Lambda_{i}^{\left(N\right)}$ are found for the sectors Toys (recreation) and Softw (computer software), and they also have the second and third highest values of $\lambda_{i}$, at $\lambda_{i}\approx 0.47$ for both sectors. These sectors not only provide a strong rotational contribution to the total observed causality flow ($\Lambda_{i}^{\left(N\right)}$) but also experience an almost equally strong effect of the gradient and rotational flows ($\lambda_{i}$) on their own dynamics. This makes them interesting candidates for future research to understand the circular dependencies in the financial network better.

### 3.3. The 2007 Financial Crisis

Because the Granger network for 2007 does not have a connectivity of 100%, it will not be analyzed as deeply as the 2020 network in this manuscript. But since only a single sector (FabPr, fabricated products) is disconnected from the rest, it might be justified to briefly focus on the reduced network of the 48 connected sectors, not least because this period also coincides with the onset of the financial crisis in the late 2000s [38] and serves as an interesting comparison to the COVID crisis. This reduced network is depicted in Figure 7, and visual inspection shows a much more streamlined flow than for the network during 2020 and hence a more hierarchically organized potential: while the 2020 network is much more entangled, the 2007 network mostly consists of links from the sector Gold to other sectors, with much fewer links between the other sectors, resulting in a shape reminiscent of the depictions of Aton in ancient Egyptian artworks. This is confirmed by the estimation of the gradient flow contribution for the reduced network, which yields $\gamma^{\left(2007\right)}=0.98$ and shows an even higher gradient contribution than that in any of the 2020 networks. Although the crisis starting in 2007 is generally known as the global financial crisis, the financial sectors do not have a particularly important position in the hierarchy of Granger causality during this period and perhaps act as mediators of causality rather than as causal drivers. This surprising result might indicate that the causality analysis for this period does not fully represent the processes in the real economy but uncovers more subtle relationships between the time series.

## 4. Discussion

Analysis of the disjoint annual periods in Figure 3 indicates that a highly connected Granger causality network coincides with market crises. The highest connectivity values, close to 100%, were observed during the 2020 COVID crisis and the beginning of the financial crisis in 2007. Other spikes occurred during the years 2023, possibly reflecting the collapse of several mid-size banks and the threat of contagion in the US banking crisis [39], and 2016, following the market turmoil after the unexpected election of Donald Trump [40]. These results align with the causality estimation in [20], as well as with research on financial correlation matrices, which also shows an increase in coupling between financial time series during times of crises [41,42,43]. The lower coupling during periods of a healthy market reflects that the time series are more independent and diversified, which reduces the overall risk in the market.

The precious metal sector is usually found upstream at the top of the Granger causality hierarchy but because of its own lack of causal drivers rather than because of the influence it exerts on other sectors necessarily. This is, to some extent, in line with the clustering analysis in [44], which identified precious metal mining companies as having different dynamics compared to other assets. Deeper insights into the estimated $CGC_{X\to Y}$ values between the sectors show that the precious metal sector is, however, not typically the strongest driver of market dynamics. Its position at the top of the hierarchy instead reflects that this sector is rarely driven by other sectors. Hence, this result should be interpreted with caution. This might be a specific feature of the precious metal sector because the pharmaceutical sector is also frequently found at the top of the hierarchy but has a strong outflux of Granger causality and therefore acts as a driving force of the system’s dynamics.Note that the returns of this sector are calculated based on companies which trade precious metals and do not directly contain the prices of gold and other metals. Adding this consideration might be an interesting endeavor for future work. During the COVID crisis, the high position of Drugs in the causality hierarchy and the rise of MedEq during the pandemic’s peak reflect our intuition about the economy during the year 2020. Perhaps surprisingly, financial sectors do not have a high position in the hierarchy of Granger causality, and especially, the banking sector is found rather far downstream. This might be interpreted as financial companies only being mediators of causal influence and providing the infrastructure for the flow of causality in financial markets but not actually driving this flow themselves. Our results therefore differ from the study in [45], where an analysis of the input–output network of business sectors showed that the energy and finance sectors have a high upstream position in the hierarchy. This is an important indicator that the financial market network analyzed in our study does not simply resemble the real economy but has its own dynamical behaviour. The high signal-to-noise ratio of the full regression models in the RCGCI algorithm and the strong contribution γ≫λ of the gradient flow indicate that the hierarchy estimated by the HHKD is reflective of the true underlying structure of the market dynamics. Interestingly, γ was notably higher for the 2007 financial crisis than for the COVID crisis, possibly because the former was an endogenous crisis and the latter an external shock. Even though the inherent uncertainty of the estimated $CGC_{X\to Y}$ values should bias the HHKD towards a more cyclical result [32], this bias seems to have been compensated for by the sparsity of the RCGCI algorithm. We hence conclude that the resulting cyclical component estimated by the RCGCI-HHKD pipeline is indeed a structural component and not merely noise.

Numerous extensions can be made in future work to this project. Adding return time series of precious metal prices has already been suggested, but beyond this, macroeconomic variables like the inflation rate could be used as background variables *Z* in Equation (Equation 1). Without attempting to create regression models to predict *Z*, these variables can still be used to calculate the Granger causality conditional based on the macroeconomic information provided by them. This might be interpreted as the third translational component that is usually omitted in Helmholtz–Hodge considerations but represents an influx into or outflux out of the whole system of interest, as discussed in [12]. Moreover, the linear regression could be extended with interaction terms between two variables $X_{t-\tau }^{\left(i\right)}\cdot X_{t-\tau }^{\left(j\right)}$ or nonlinear functions [46] to alleviate the shortcomings of Granger causality methods [47,48], but this might require larger amounts of data for reliable estimation and thus higher-frequency data than those available in [22]. Other methods from causal inference, such as the lead–lag relationship of complex Hilbert PCA [49,50] or transfer entropy [7,19], can capture nonlinear effects but might require more data, too. Although the heightened position of precious metals and pharmaceutical products can be related to real effects in the data, the restriction of Granger causality as a linear measure may have been the reason why financial companies do not have a high position in the estimated hierarchy of causation. As this is rather counter-intuitive, nonlinear causality measures might be used to provide a different perspective on the data and to check the robustness of these findings. Also, one could extend the RCGCI algorithm to include a bootstrapping procedure in the estimation of (Equation 3) to obtain an estimation of the uncertainty of the Granger causality $CGC_{X\to Y}$. While the RCGCI and the standard formulation of Granger causality do not distinguish between positive and negative influences between variables, similar to [21], a multi-layer network approach could be used to separate the causal couplings based on their sign. However, extending the HHKD to multi-layer networks is required to evaluate this, perhaps based on the approach in [51].

Because of the high network connectivity during crises, the RCGCI-HHKD pipeline is especially useful for describing the system dynamics during such periods. In particular, understanding the flow of causality and identifying the causal drivers during a crisis might allow policymakers to more effectively intervene to stop crises by focusing on the sectors which are upstream in the causality hierarchy. This could open up the possibility of stabilizing the market with a minimally invasive intervention.

Finally, though this is beyond the scope of this work, we believe that the HHKD could help to overcome the limitations of the causality framework described by Judea Pearl [3]. Pearl’s approach to causality relies on directed acyclic graphs (DAGs) between the variables and therefore requires an interaction network without any closed loops. While this is not always present in real systems, an adaptation of the HHKD might provide a suitable tool for extracting such DAGs from real-world systems as the gradient component of the original graph.

## Figures and Tables

**Figure 1 entropy-26-00858-f001:**
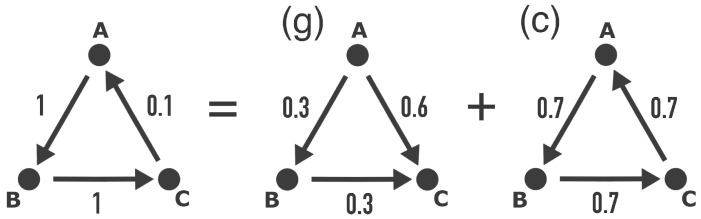
Example of the Helmholtz–Hodge–Kodaira decomposition for a single graph into a gradient-based graph (g) and a circular graph (c). Note that direction of the flux between A and C is different in (g) and (c), which is the same as changing the sign $J_{AC}^{\left(g\right)}=-J_{CA}^{\left(g\right)}$, and hence, their sum is given by $J_{CA}^{\left(g\right)}+J_{CA}^{\left(c\right)}=-0.6+0.7=0.1$, and the original flux $J_{CA}$ is reconstructed. Also, note that the total flux between two nodes is path-independent for (g) as $J_{AC}^{\left(g\right)}=J_{AB}^{\left(g\right)}+J_{BC}^{\left(g\right)}$.

**Figure 2 entropy-26-00858-f002:**
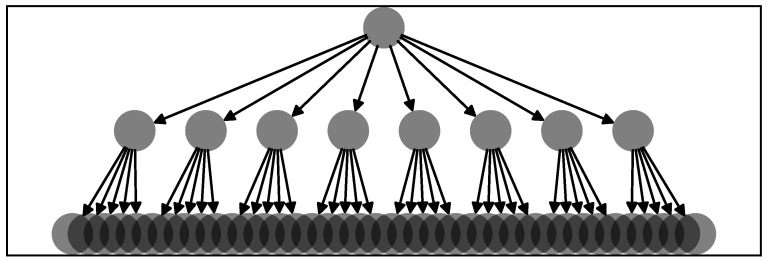
The network structure used for the vector autoregression which generates synthetic time series. One node is at the top of the hierarchy without any causal parent, whereas eight nodes are in the second layer and forty are in the final layer. Each node in the second layer is the parent node of 5 nodes in the final layer and has the node in the first layer as their parent node. Sketched via the software [35].

**Figure 3 entropy-26-00858-f003:**
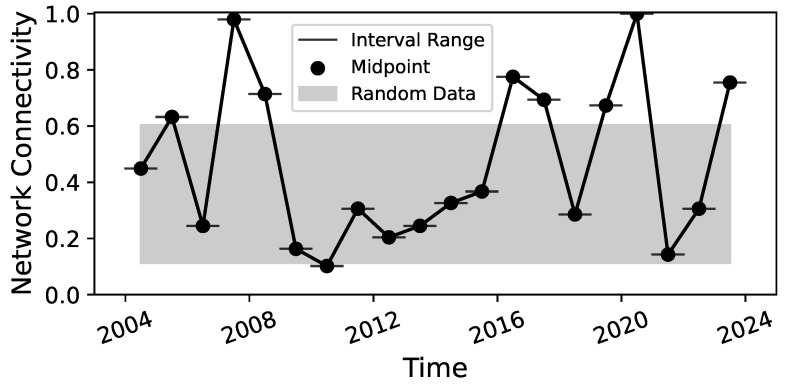
Results of the RCGCI-HHKD analysis for annual data from [22]. The gray shaded area is the CI for the network connectivity of random data without any causal coupling. Note that the lines that connect the dots are only a visual aid, and no linear interpolation between the periods is assumed.

**Figure 4 entropy-26-00858-f004:**
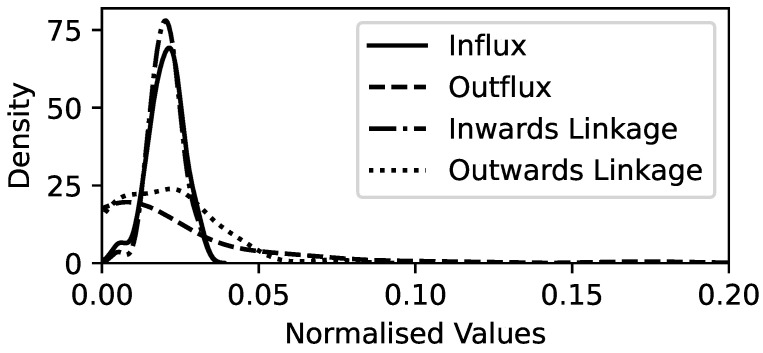
For the analysis of annual data from 2004 to 2023, KDE of the sum of all influx and outflux of Granger causality and the total number of years with at least one inward or outward link in the RCGCI network. Values on the x-axis have been normalized to the same scale.

**Figure 5 entropy-26-00858-f005:**
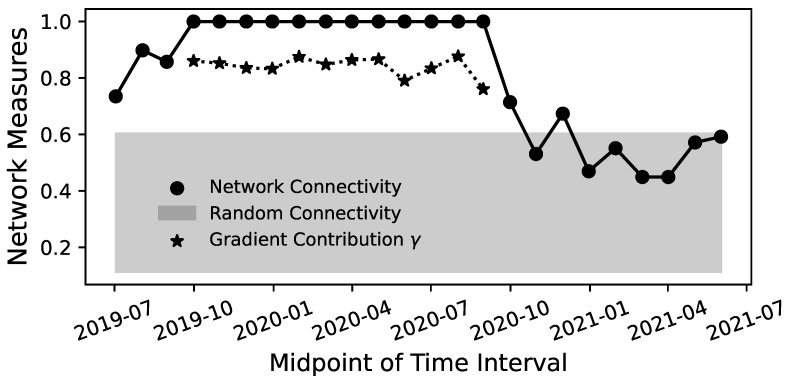
For time periods of 12 months, two network measures are depicted here: the network connectivity and the gradient contribution γ. The network connectivity is the percentage of sectors connected to the network and is displayed here against the random connectivity expected for independent time series. If the network is complete and has a connectivity of 1, the gradient contribution γ is also calculated according to Equation (Equation 10). Note that the time on the x-axis is the midpoint of the 12-month intervals of data.

**Figure 6 entropy-26-00858-f006:**
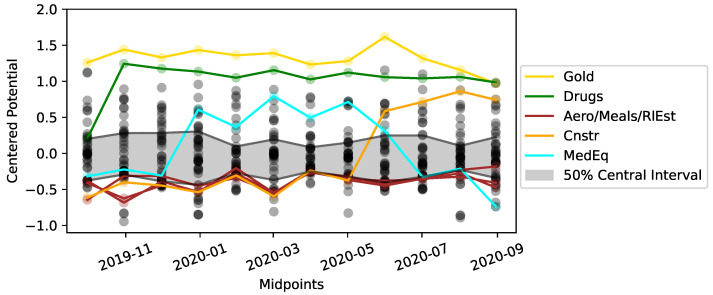
For the same time intervals as in Figure 5, the potentials $\Phi_{i}$ of each sector are shown as dots. Note that for each time interval, the potentials have been centered via $\sum_{i}\Phi_{i}=0$. Some selected sectors are shown in color, and the gray area shows the spread between the 25% and 75% quantiles for each time period.

**Figure 7 entropy-26-00858-f007:**
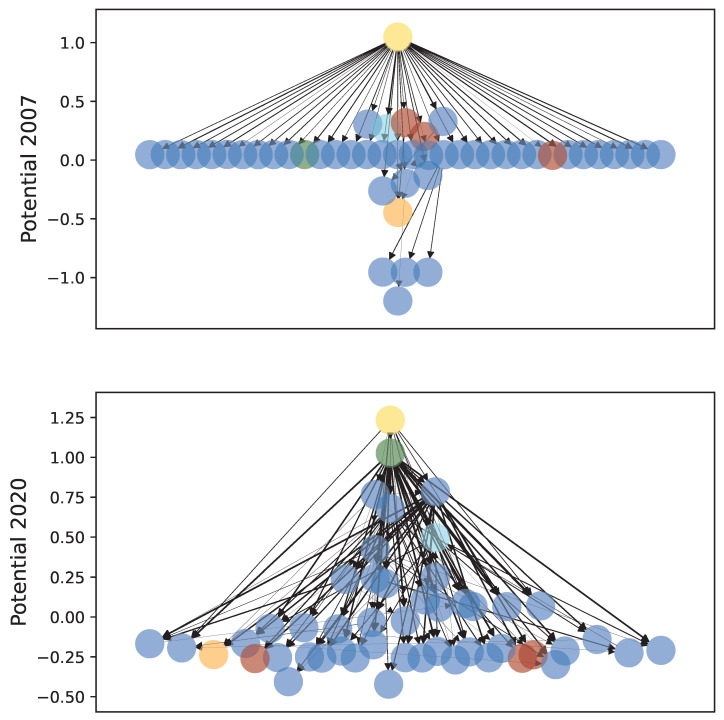
The estimated Granger causality influence network ordered by the HHKD potentials for the periods from January 2007 to December 2007 (the sector FabPr is not shown because it has no link to any other sector) and from October 2019 to September 2020. The width of the arrows reflects the strength of the Granger causality, and selected sectors are highlighted with the same color coding as in Figure 6 whereas all other sectors are shown in blue.

## Data Availability

The data used for this research project, as well as the main code for the RCGCI-HHKD pipeline, are published at [52] in Zenodo.

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
