# Peer review of "Causal Hierarchy in the Financial Market Network—Uncovered by the Helmholtz–Hodge–Kodaira Decomposition"

_entropy, 2024, doi:10.3390/e26100858_

Round 1

Reviewer 1 Report

Comments and Suggestions for Authors

I believe that the results presented in this article can make a significant contribution to understanding the sometimes non-obvious, complex relationships between different markets.

Author Response

We would like to thank Reviewer #1 for their very positive and timely feedback on our submission.

Reviewer 2 Report

Comments and Suggestions for Authors

The authors present and discuss methods of cross-correlation analysis which take into account the causality of events employing the Granger methodology and combining it with the network decomposition into rotational and gradient components.

Generally speaking, the method proposed seems to be very promising however it should be better justified. It is not clear why the proposed decomposition could increase our knowledge or predictability of the system.

Apart from the main problem, there are several misunderstandings which could be easily corrected.

1.  The return does not ensure stationarity, so this sentence should be corrected.

2. l.58 Here model is fitted. IN AI the models are trained, but it is a different approach.

3. The network theory terms are used embarrassingly.

a) "complete network" -> fully connected, or just "connected"

b) "perfect connectivity" ? this term is unclear, the same connectivity should be defined because in the literature it can be defined variously.

c) fig. 3 the lines connecting the points do not make any sense. I am absolutely sure that extended analysis i.e. at periods within this year would not give linear change of the connectivity.

d) l. 220 "normalization" should be defiend, the same "scale"

e) Instada describing PDF as "resembling" is much better to apply any of the tests,

f) the PDF of the sectors are not presented so the reader has not chance to verify if they "resemble" something.

g) l 234 "completely connected" means "connected" or fully connected?

h) description of figure 5 "network measure"?

i) l. 245 sentence unclear "strongly depends" - it is not defined

j) fig 6 What is the meaning of the dots?

k) Is the potential 2007 different from 2020?

Comments on the Quality of English Language

Problem with the network vocabulary usage.

Reviewer 3 Report

Comments and Suggestions for Authors

The authors apply the Hodge-Helmholtz-Kodiara decomposition to effective causality networks constructed by applying a Granger causality estimator to financial time series data. The methodology presented here is interesting and could be of interest to researchers in other domains who study interacting time series organized by an underlying network (e.g. neuroscience). 

The presentation is clear and the figures are all legible. The mathematics is presented briefly, but is, for the most part, sufficiently clear. The bidirectional extension of the HHKD could be explained more clearly (lines 143 - 160). A more serious discussion of the representational biases of the HHKD would also improve the paper. The paper also lacks any attempt to quantify the uncertainty inherent in the causality estimates, which propagate to uncertainty in the HHKD characterization. 

Commentary:

1. The HHKD extracts a transitive component by solving a weighted least-squares problem. While minimizing a square error is standard practice, it has particular consequences on the potentials produced. Namely the potentials generated aim more to minimize large errors than small errors, and to distribute any necessary errors across many edges. This produces a circulating component that is highly distributed. It also treats large causal relations as more informative than small relations. This is reasonable, but should be accounted for statistically by weighting the decomposition according to the uncertainty in the estimated causality. 

2. The description of the synthetic validation experiments is difficult to follow, and it is unclear how it justifies later expectations (e.g. the expectation referenced in line 210).

3. The causal quantities used to build the decomposed networks are estimated from sample data assuming a stochastic underlying model. These are estimators applied to data that is, under the assumed model, a random sample. Thus, the estimators are themselves random variables, and will include sampling error. The noise in the estimated quantities is not discussed anywhere in the paper. This is a problem since uncertainty in the causal quantities will produce uncertainty in the downstream analysis. For example analyses see Reference 12 and the following paper (Strang, Abbott, Thomas. The network HHD: Quantifying cyclic competition in trait-performance models of tournaments. SIREV. 2022.) This is an important issue since uncertainty in the estimated causal network biases the estimated HHKD characterization, and the degree of bias depends on the density of the network.

4. It is unclear whether the authors consider circulation as necessarily noise, or as a structural component of the causality network. This point must be clarified, especially in the discussion where the authors reference Pearl.

5. The paper should include a more critical discussion of the methods presented. In the authors' opinion, did the HHKD return a meaningful summary of the causal network?

Comments on the Quality of English Language

The paper is largely well-written. Outside of occasional typos (line 120 "fulfils" instead of "fulfills", line 152 "minimisation" instead of "minimization") the paper does not include many specific writing errors. I

Round 2

Reviewer 2 Report

Comments and Suggestions for Authors

Generally speaking, most of the raised issues have been properly answered.

However, the problem of "stationarity" requires explanations.

I'm afraid I have to disagree that the return affects stationarity. To find proper explanations of the return transformation I can mention the most typical reasons:

1. the return normalizes the time series. The shares can be of different levels and even units.  The return converts it into percentage values. So comparable.

2. The second reason is that it removes trends caused by random walks.

3. The random walk is often used to model financial time series and the return is used to verify the idea.

4. It is also important to be aware that there is also logarithmic return which is often used in high-frequency time series. Of course, there are some reasons to apply the logarithm return. But I suggest verifying it in appropriate literature...
